# Effect of Flow Field Optimization of an Asymmetric Multi-Strand Tundish on the Quality Consistency of Cracking Con-Rod Steel

**DOI:** 10.3390/ma15103698

**Published:** 2022-05-21

**Authors:** Zhanpeng Tie, Haiyan Tang, Kaimin Wang, Hongsheng Miao, Sen Cai, Fenqiang Xian, Jiaquan Zhang

**Affiliations:** 1School of Metallurgical and Ecological Engineering, University of Science and Technology Beijing, No. 30 Xueyuan Road, Haidian District, Beijing 100083, China; tzp15003685525@163.com (Z.T.); wangkaimin0906@163.com (K.W.); 15901131365@163.com (S.C.); 2Technology & Quality Center, Xining Special Steel Co., Ltd., 52 Chaidamu West Road, Xining 810005, China; mhs1972@126.com (H.M.); xfq1456322319@163.com (F.X.)

**Keywords:** high carbon steel, cracking con-rod, C70S6, tundish, flow field, water modelling, mathematical simulation, temperature field, bloom quality consistency

## Abstract

Cracking con-rod is an advanced high-precision connecting structure based on brittle expansion, breaking and reconnection of steel, to solve the problem of assembly circle missing. High carbon micro-alloyed steel C70S6, as a dominant material for the production of cracking con-rod, has extremely strict requirements on non-metallic inclusions in steel and microstructure stability. Continuous casting tundish plays an important role in removing large-sized inclusions and stabilizing casting quality. Aiming at the inconsistent casting quality of C70S6 steel produced by a three-strand asymmetric tundish and the frequent occurrence of slag entrapment problems in Xining Special Steel, the tundish structure was optimized by means of physical modelling combined with numerical simulation, and the quality of the bloom castings and subsequent hot-rolled products before and after optimization were compared based on volume production. The results show that a new flow control design to the tundish can effectively improve the consistency of its metallurgical effect for each of the three strands and the following overall product quality, in which the flow field and temperature field in the tundish are more uniform. This is due to the adoption of a vortex inhibitor and an optimized wall structure according to the measured RTD curve, ink trajectory and numerical simulation on the 3-D streamline contours and temperature distribution in the tundish. The peak concentration of outlet 1 is decreased from 6.5 before optimization to less than 2.0 after optimization, which means the elimination or alleviation of the local short-circuit flow. The maximum temperature difference of C70S6 molten steel measured at the outlets of the tundish three strands is decreased from 2–5 °C to 1–3 °C, which is in good agreement with the numerical simulation results. The difference in columnar crystal ratio of the corresponding bloom castings is decreased from 2.27–3.17% to 1.26–1.85%, and the consistency of central carbon segregation index is also significantly improved. In addition, the difference in oxygen content among the three strand blooms is decreased from 1.7–3.5 ppm to 0.8–1.9 ppm. As a result, the overall mechanical properties and microstructure stability of the hot-rolled products are improved statistically, in which the hardness fluctuation is decreased from 84 HBW to 60 HBW, the inclusion grade of types B + C + D + Ds is reduced to 1.105, and the occurrence rate of Ds dropped to 0.118%. Accordingly, the failure rate of the cracking con-rod is controlled stably within 4‰, and the fracture is generally smoother than that before tundish optimization. In summary, the flow field optimization to a multi-strand asymmetric tundish has a clear effect on improving the overall quality of its bloom castings and rolled products, which should be paid more attention industrially. Meanwhile, the present study provides a reliable theoretical and experimental reference for the improvement of metallurgical effects of an asymmetric-typed tundish commonly used in special steel production.

## 1. Introduction

Cracking con-rods are important transmission components connecting crankshafts and their pistons, which bear high requirements on load capacity, installation and service precision. As an advanced connecting structure, a cracking con-rod is made based on brittle expansion, breaking and reconnection, which is expected to solve the frequent issue of assembly circle missing. To produce a precise cleavage fracture for tight joints and avoid big head distortion of the reconnecting rod due to plastic deformation, very strict demands for a uniform microstructure of steels needs to be met [1]. The non-quenched and tempered free cutting steel C70S6 is a near eutectoid micro-alloyed steel invented in Germany for cracking con-rod manufacture [2]. It is also a new type of energy-saving and efficient steel, with features of high-carbon, micro-alloyed, high-sulfur content and narrow chemical composition range, thus it is relatively difficult to produce by continuous casting. In addition, non-metallic inclusions in steel destroy the continuity of the steel matrix, causing stress concentration and inducing crack formation, thus improving the cleanliness of molten steel is one of the key demands to solve this problem. 

From the perspective of structure and performance requirements on finished products, on the one hand, the high-carbon and high-sulfur steel has a large tendency to shrink and crack in the process of continuous casting, which will easily result in the failure of ultrasonic detection. Additionally, its large segregation tendency due to its inherent chemistry can easily cause poor stability of the microstructures and mechanical properties of the subsequent rolling products [3,4,5]. Therefore, an in-depth understanding of the as-cast steel microstructure and its defect characteristics are of great significance on developing a continuous casting practice of this type of high-class steel. For this reason, we systematically characterized its solidification microstructures and composition segregation characteristics in the present continuous casting condition based on industrial development tests [6]. In this study, the as-cast genetic defects which may affect the requirements of the expanding cleavage fracture during the processing of steel products were revealed, and the technological approaches of optimizing continuous casting process were put forward. As a result, the quality of the bloom castings has been improved to an applicable degree of the users. However, due to the asymmetric three-strand tundish used in this steel plant, the temperature difference is large among different strands in tundish caused by an uneven flow field, in which the maximum temperature difference of molten steel measured at the outlets near the stopper reached 5 °C. Moreover, the bloom qualities of C70S6 steel and subsequent rolled products corresponding to each strand are not at a consistent level. Therefore, it is of great necessity to control the consistency and stability of casting conditions from tundish, at the very beginning of continuous casting.

Tundish, a transition container connecting ladle and mold in the casting process of molten steel, not only plays the role of receiving liquid steel and distributing it to the molds, but also has the metallurgical functions of homogenizing the composition and temperature of liquid steel and removal of non-metallic inclusions [7,8,9]. For the continuous casting of high-quality special steel, the consistency of the metallurgical effect of each strand in the tundish is key to improving and stabilizing the overall quality of subsequent products, representing a popular issue in recent years [10,11]. A reasonable tundish structure can ensure a good flow-state of liquid steel in the tundish, which is beneficial to improving the consistency of each strand, reducing the temperature difference between tundish outlets, and prolonging the residence time of liquid steel in the tundish, thus promoting the removal of inclusions.

In view of the high temperature characteristics of continuous casting tundish, the optimization of its structure is mainly carried out by water modelling [12,13,14] and numerical simulation [15,16,17]. For example, Tang et al. [18] optimized a 4-strand tundish in a steel mill through the above method. After application in the steel production, it was found that macroinclusions and microinclusions in the billet castings were reduced by 44.9% and 2.7%, respectively, compared with the use of the prototype tundish, and the total oxygen and inclusion contents in the four-strand billet castings were very consistent. Chen et al. [19] used the same method to optimize the flow field of a two-strand symmetrical tundish for slab production. By increasing the distance between dam and retaining wall, the average residence time of water in the simulated tundish and volume fraction of the piston zone were increased, and the fraction of dead zone was reduced. The total oxygen content of billet was reduced by 30.32% after applying the optimized tundish in production. Merder et al. [20] studied the motion behavior of non-metallic inclusions in a two-strand tundish by using different simulation models and showed that the process of microparticle distribution can be well described by the Euler–Lagrange method according to the DPM model. In addition, their distribution characteristics and local concentrations can be reflected with a high accuracy. Sheng et al. [21] applied a CFD model to study the effect of thermal buoyancy on fluid flow and residence time distribution in a single-strand tundish. The results showed that thermal buoyancy has a significant impact on the flow pattern and temperature distributions of molten steel in the tundish. The increase of heat loss through the top surface shortens the mean residence time of molten steel in the tundish, and thus leads to an increase in dead volume fraction and a decrease in plug-flow volume fraction. Sheng [22] also investigated certain key aspects regarding the CFD simulation of a tundish, such as the choice of turbulence model, meshing, boundary conditions and a discretization scheme. In his study, a mesh size of 0.006 m, realizable k-ε turbulence model and second-order upwind discretization scheme are the recommended modeling settings for tundish flow simulation. 

In the present paper, a three-strand continuous casting tundish was studied for continuous casting of cracking con-rod steel in Xining Special Steel. The tundish has an asymmetric geometric structure with a casting position of the ladle long nozzle close to its No.1 and 2 strand, but far from the No. 3 strand. Usually, an asymmetric tundish is more difficult to design than a symmetric tundish, and the uniformity of the flow field is also poorer. The industrial big data found that under the prototype flow control device of the tundish, the molten steel temperature at the first strand outlet and the occurrence rate of inclusions in the final cast products were higher, while its equiaxed crystal ratio measured lower than the other strands. Although there are many reports on the optimization of tundishes, most studies focus on symmetric multi-strand tundishes, which are evidently different from their asymmetric counterparts.

To meet the strict requirements of the cracking con-rod regarding the consistencies of the microstructure and properties of the rolled products, the tundish structure is optimized by physical and numerical simulation methods in the present study, and an overall quality comparison of the bloom castings and the following hot-rolled products before and after optimization is given based on a large number of industrial tests. These on-site volume data will be of great value in evaluating the casting practice for the quality production of special steel. Meanwhile, this study intends to provide a roadmap or benchmark for the optimization of other asymmetric multi-strand tundishes.

## 2. Production Process and Basic Parameters of C70S6 Steel

The manufacturing process of C70S6 steel is as follows: 110 t Consteel EAF (electric arc furnace) → 75 t LF (ladle furnace) + 75 t VD (Vacuum degas) → three-strand 250 mm × 280 mm bloom continuous caster → hot rolling → magnetic plus ultrasonic flaw detection. The chemical composition is shown in Table 1, indicating that this steel has the characteristics of high C, S and N content, and low Si and Al content.

Steel is cast by a three-strand tundish whose structure and original flow control device is shown in Figure 1. The flow of molten steel is mainly controlled through a retaining wall with a diversion hole measuring 140 mm in diameter and angled 8° upwards on every side wall. After the molten steel is poured into the impact zone from a long nozzle, it is transferred through the diversion hole in the side wall to three submerged entry nozzles (SEN). The process parameters of the continuous casting of C70S6 steel are shown in Table 2. 

## 3. Research Methods

### 3.1. Physical Modelling

Taking the industrial tundish shown in Figure 1 as the prototype, a model tundish with a geometric similarity scale of 1:2 was designed based on the similarity principle, and water was used to simulate the flow of molten steel in the tundish. According to the similarity principle, the Froude (*Fr*) numbers of the prototype and model should be equal [13], which can be expressed as: (1)$$Fr_{m}=Fr_{p}$$
where m denotes the tundish model, and p is the prototype.

The *Fr* is defined as: (2)$$Fr=\frac{u^{2}}{gL}$$
where *u* represents flow velocity, *L* is characteristic length of tundish and *g* is gravity acceleration. 

Thus, the main geometrical and operation parameters between the model and prototype can be expressed with similarity ratio *λ* as follows: (3)$$L_{m}/L_{p}=\lambda ,\;V_{m}/V_{p}=\lambda^{3},\;u_{m}/u_{p}=\lambda^{0.5},\;Q_{m}/Q_{p}=\lambda^{2.5}$$
where *V* is tundish volume and *Q* is the volumetric flowrate of molten steel.

Substituting in the data from Table 2, *Q*_m_ and *V*_m_ can be obtained, respectively.

The stimulation–response method [13,23] was adopted in this experiment. When the liquid level of tundish reached a given height, 100 mL saturated KCl solution was injected as a pulse tracer from a side port mounted at the upper position of the ladle shroud. The conductivity of KCl at the three outlets of tundish was monitored by three conductivity probes. The conductivity curve with time (RTD curve) was obtained by DJ800 (IWHR, Beijing, China) system. 

The stagnation time *t*_min_ and peak time *t*_peak_ of the fluid in the tundish was directly read from the RTD curve. Other flow characteristic parameters were obtained according to the modified mixing model proposed by Sahai Y [24], as shown in Table 3, where *C(t)* is the measured tracer concentration at the time *t*; *j* is a specific strand, *j* = 1, 2, 3; *n* is total strand number, *n* = 3. The optimization effect of flow control device was evaluated by analyzing the flow characteristic parameters of molten steel in tundish and the RTD curve.

To visually display the flowing trajectory of molten steel from ladle to mold, ink was added to the water in the tundish from the same position where KCl was injected, and its dispersion was recorded by a camera.

### 3.2. Mathematical Model

A mathematical model including four main governing equations was developed to calculate the flow field and temperature profiles of molten steel in the prototype and optimized tundishes, with the casting speed set as 0.75 m/min for a bloom section of 250 mm × 280 mm. These equations and their designations are given in Table 4, and details of the model are described in refs. [10,13,25].

### 3.3. Assumptions and Boundary Conditions

#### 3.3.1. Main Assumptions

The following assumptions were made for simulation calculation: liquid steel in tundish was incompressible single-phase turbulent flow; influence of the slag layer on flow was ignored; the slag-steel surface was treated as a free surface. 

#### 3.3.2. Boundary Conditions and Solution Methods

The inlet of the molten steel in tundish adopted a velocity inlet. The inlet temperature was set as 1773 K based on the casting temperature of steel C60S6. The direction of speed was perpendicular to the section of the long nozzle inlet, and its value was obtained by the size of the casting section and the casting speed according to the mass conservation, which is 0.486 m/s in the present study. The tundish outlet adopted outflow boundary conditions, the same casting speed for each strand; the melt surface was considered to be flat, and the vertical gradient of all variables on the free surface was zero; the tundish wall surface was regarded as a non-slipping condition, and the standard wall function was adopted to deal with the near wall; the heat dissipation of each wall of the tundish adopted the second type of boundary condition. The material parameters and boundary conditions of the model are shown in Table 5, in which the turbulence kinetic energy and turbulence energy dissipation rate were obtained by semi-empirical formulas $k=0.01v^{2}$ and ${\varepsilon =2\;\times \;k}^{1.5}/D$, respectively, (*v* represents the flow rate of molten steel and *D* represents the diameter of the long nozzle), and the others originated from reference studies [10,16]. 

The flow and temperature distribution of molten steel in the tundish were simulated by commercial CFD software FLUENT 14.5 (ANSYS, Pittsburgh, PA, USA, 2012). Choosing the pressure-based solver for calculation, and the pressure and velocity coupling were based on the SIMPLE algorithm. The computational convergence residual was set to less than 10^−6^ for the energy equation and 10^−4^ for the other equations.

## 4. Experimental Results and Discussion

### 4.1. Results of Water Modelling

In total, 30 groups of experimental schemes were designed and carried out, but only the tundish prototype and the final optimization scheme are relevant to the comparative quality analysis here, thus this paper only lists the two schemes for comparison. The retaining wall structure of prototype case A was shown in Figure 1. The optimized case B is shown in Figure 2, in which a diversion hole measuring 90 mm in diameter and upward angle of 10° is displayed on the right side of the wall near outlet 1, and two diversion holes are displayed on the left side near outlet 2, with each hole at the same diameter and upward angle as on the right side but a different horizontal inclination of 3° and 5°, respectively. Additionally, a vortex inhibitor is designed as shown in Figure 2b. The flow characteristic parameters of the two schemes are shown in Table 6.

As can be seen from Table 6, the proportion of dead zone (*θ*_d_) in the tundish prototype case A reaches as high as 53.08%, while the piston zone *θ*_p_ is only 2.38%. A large proportion of dead zone indicates that molten steel flows much slowly in some areas of tundish, which renders present non-metallic inclusions difficult to collide and grow, and the local molten steel temperature will also be relatively low due to weak heat exchange. The stagnation times *t*_min_ of outlets 1 and 2 are 17 and 16 s, respectively, and the peak time *t*_peak_ is 20 and 19 s, respectively. The difference between *t*_min_ and *t*_peak_ is only 3 s, suggesting that the short-circuit flow is possibly formed at the two outlets. The stagnation time of outlet 3 is about 2.5 times that of outlets 1 and 2, and its average residence time is much longer than that of the other two, indicating poor consistency among strands. This is because the diversion holes are far from the No. 3 strand, while pointing to Nos. 1 and 2 in direction, and the fluid flows out from outlets 1 and 2 first. Its kinetic energy is small when flowing to outlet 3, resulting in larger standard deviations of the stagnation time and average residence time among the three strands, *S*_tmin_ and *S*_tav_, are 19.20 s and 25.56 s, respectively.

Compared with case A, the dead zone proportion of case B is only 21.1%, and the stagnation times of outlets 1 and 2 are extended to 30 and 31 s, respectively. The standard deviations of stagnation time and average residence time are reduced to 11.12 and 15.22 s, respectively, indicating a clear improvement of the consistency among strands.

Figure 3 shows the RTD curves of the two schemes, where the ordinate is dimensionless concentration and the abscissa is dimensionless time. As can be observed, after the tracer KCl is added into the tundish in case A, the concentrations of outlets 1 and 2 rapidly reach the peak value, and the peak concentration of outlet 1 is as high as 6.5, indicating that a short-circuit flow has formed in the tundish. The short-circuit flow will make the molten steel flow out of the tundish rapidly. In addition, its residence time in the tundish is short, and there is inadequate time for inclusions to grow or float. This may be related to the lower pass rate of ultrasonic detection of the rolled products corresponding to tundish outlet 1 in industrial production.

Although the RTD curve of case B is not very smooth, the peak concentration is greatly reduced to less than 2.0, and the consistency among different strands and the short-circuit flow are also improved as compared with case A. In addition, the average residence time of the three strands ranges from 591 to 651 s, much longer than the 267 to 434 s in the original case. This will provide enough time for inclusions in the molten steel to float and be removed, and thus lead to an improvement in the cleanliness of the steel.

Figure 4 compares the ink trajectory of the two cases. The ink tracer consists of 50 mL black ink mixed into 150 mL water. From Figure 4a, it can be seen that the ink from diversion holes in case A flows downwards and reaches outlets 1 and 2 at about 16 s, then reaches outlet 3 at about 60 s. This is basically in agreement with the experimental measurement. The ink moves to the side wall of the tundish and then flows back to the middle area. The top region of outlet 2 mixes slowest, possibly representing the main dead zone in the tundish. The local temperature of molten steel will possibly also be low. The ink fills the whole tundish at about 180 s in this case. By comparison, the ink in case B (Figure 4b) flows upwards due clearly to the increased upward inclination angle of the diversion holes from 8° to 10°, with which the fluid will be delayed in arriving at outlets 1 and 2. However, because of the reduced hole size (from 140 mm to 90 mm) in the wall, the fluid velocity will increase, which accordingly shortens the stagnation time of outlet 3 to 51 s. As a result, the flow of molten steel will become more uniform in case B than in case A. The ink is completely mixed at 124 s, 56 s shorter than that in case A. 

### 4.2. Validation of CFD Modelling and Mesh Independence Test

The numerical simulation method has been widely used in the structural optimization of continuous casting tundish, and the authors have also conducted an abundance of research works on this topic [10,13,18] with its reliability confirmed by water simulation experiments and industrial application results [10,18]. This paper adopts the same mathematical modeling and calculation method, and the reliability of the simulation will be further verified through actual temperature measurements. 

Meanwhile, a mesh independence test was carried out because of its importance to the simulation results. As shown in Table 7, different numbers of grids were designed for case A and case B, respectively. By applying the same boundary conditions and parameters, the temperature of each tundish outlet was calculated and compared with industrial measurement data. Data from the table indicate that the temperature of each outlet is slightly changed and the calculation accuracy can be met when the number of grids is about 300,000~600,000. Therefore, 394,471 tetrahedral grids for case A and 559,800 tetrahedral grids for case B are chosen for the following studies. 

### 4.3. Results of Numerical Simulation

Figure 5 shows the 3D streamline contours of cases A and B. It is seen that in case A, the velocity of molten steel in impact zone is great with most at 0.05 m/s. In addition, the flow is active in the area near outlet 1 as the diversion hole on one side of retaining wall points to it adjacently. The local velocity reaches over 0.02 m/s. However, the molten steel at outlets 2 and 3 flows slowly at under 0.01 m/s. This is due to deviation of the diversion hole on the right wall away from outlet 2 and further still from outlet 3. By comparison, the flow field of case B is more uniform, and the flow velocity and streamline of molten steel near outlets 2 and 3 are clearly intensified, due to the adoption of an optimized wall structure. The area with flow velocity over 0.02 m/s increases, and the velocity of the impact zone is greatly reduced with the help of a vortex inhibitor. The flow field in the tundish is crucial to the temperature uniformity of the molten steel within. 

Figure 6 compares the temperature distribution of cases A and B in the vertical section of the outlet center. Similarly, the temperature distribution in case B is more uniform than in case A. The maximum temperature difference at three outlets of tundish is only 1 K, 2 K lower than case A. The maximum temperature difference in the whole section is decreased from 8 K (1769 minus 1761 K) to 6 K (1769 minus 1763 K), and the high (1768~1769 K) and low temperature areas (1761~1763 K) are both also reduced in case B.

From the results given above, the consistency of flow and temperature distribution of case B are both significantly improved compared with case A, which is expected to improve the stability and consistency of steel product quality among different strands. 

## 5. Comparison of Industrial Application Effect before and after Tundish Optimization

Based on the results of the physical and numerical simulation, the optimized case B was successfully adopted to industrial production of C70S6 steel. The temperature difference of molten steel at the three outlets of the tundish, and the qualities of bloom castings and subsequent rolled products before and after optimization were compared under the same given refractory material and operating conditions. 

### 5.1. Comparison of Temperature Difference of Molten Steel

The molten steel temperature in the tundish was measured comparatively before and after tundish optimization, in which the 2nd, 3rd, 4th, 5th and 7th heats were selected for each casting test, three times for each heat during the casting process. A temperature detector (TC1200-S-AL, Heraeus, Hanau, Germany) was inserted into the molten steel from the stopper holes between strands, and the maximum temperature difference among the three strands was recorded. For better comparison, measurement of the three strands was carried out at nearly the same casting time. There are 13 groups of effective data obtained for the comparison, which are shown in the Figure 7. 

As is seen, the maximum temperature difference between strands measures 2–5 °C before tundish optimization, and the average temperature difference is 3.2 °C. After optimization, the maximum temperature difference decreases to 1–3 °C, and the average temperature difference drops to 1.8 °C, indicating an improved temperature field uniformity of the tundish. Additionally, the maximum temperature difference calculated by numerical simulation was 3 °C before optimization and 1 °C after optimization, which also verifies the reliability and accuracy of the numerical simulation from an industrial perspective. Meanwhile, the results also show that the flow field optimization has significantly improved regarding temperature consistency in the tundish. As we know, uniformity of the temperature field is helpful in stabilizing and unifying the superheat for the three-strand castings, and thus stabilizing the as-cast solidification structure of the strands, along with the identical quality of the as-rolled and subsequent products.

### 5.2. Comparison of Erosion of Molten Steel on Refractory Material in Tundish

The change of flow field of the tundish may affect the erosion of its lining, and thus affect the cleanliness of blooms and the following rolled products. For this reason, the eroded appearance of the tundish working layer at three positions near the slag line was compared at the end of the casting. The test positions (A, B, C) are illustrated in Figure 8. The eroded appearances of the three tundishes were examined before optimization, and five tundishes were examined at the same locations after optimization. To ensure the comparability and effectiveness of erosion data before and after optimization, the material of the working layer, amount and method of cover slag added, level height of molten steel in tundish, fluctuation of casting speed and temperature measurement times were strictly controlled identical under the two working conditions. The erosion width and depth are shown in Table 8. 

As shown in the Table 8, the erosion width of the working layer at position A close to strand No. 1 decreases from 90–120 mm before optimization to 90–100 mm after optimization, and the depth decreases from 10–12 mm to 9–11 mm. This is because a diversion hole in the retaining wall before optimization directly points to zone A, imparting significant washing to the local lining material. After optimization, the diversion hole is provided with an upward and horizontal deflection angle to avoid direct impingement on the tundish wall. This comparison indicates that a good flow field can reduce erosion of the lining, and thus hopefully reduce exogenous inclusions from erosion. The erosion widths of positions B (near No. 2 strand) and C (No. 3 strand) are basically the same before and after optimization, while the erosion depth is slightly reduced after optimization. 

### 5.3. Comparison of Bloom Quality Consistency

The effect of tundish flow field optimization on bloom quality should be embodied mainly by the improvement of bloom quality and that of its consistency among different strands. For this reason, 10 batches of C70S6 steel blooms in total were compared, with 5 batches (1#–5#), respectively, before and after tundish optimization. For each batch, No. 1 to No. 3 strand blooms at the same casting time were taken and machined, then acid etched with 1:1 HCl solution at 70 °C for 20 min. The columnar, equiaxed crystal rates and the center shrinkage were evaluated and their differences among the three strands were revealed. Moreover, the rod samples were taken from the same position of each strand bloom, and then the total oxygen content was measured with an oxygen–nitrogen analyzer (TCH600, St Joseph, MI, USA). The results are shown in Table 9. 

Table 9 shows that before flow field optimization, the columnar crystal percentage of three-strand billets distributes from 43.86–58.19%, and the difference among different strands ranges between 2.27–3.17%. After optimization, the difference of columnar crystal rates among strands is reduced to 1.26–1.85%, and the difference of equiaxed crystal rates decreases from 2.25–2.87% to 0.48–2.10%. There is no significant change in center porosity before and after optimization. It is well known that equiaxed and columnar crystal proportions are closely related to casting superheat. A high superheat means a greater temperature gradient between the inner liquid core and the outer, along with a lower solidification nucleation rate, and thus resulting in a more developed columnar crystal. Meanwhile, the excessive growth of columnar crystals causes the enrichment of solute elements in the center, generating center segregation. Before tundish optimization, the temperature of molten steel near No.1 strand is higher than that of other two strands, thus the superheat is relatively high, and accordingly the proportion of columnar crystals is slightly higher than that of other strands from statistics. After optimization, the temperature field is more homogeneous, thus the microstructure difference among strands is reduced.

Moreover, Table 9 shows that the oxygen content difference between three strands of blooms is 0.8–1.9 ppm after flow field optimization, while it is 1.7–3.5 ppm before that, indicating that the consistency of the bloom cleanliness has been effectively improved. Meanwhile, the oxygen content in the No. 1, No. 2 and No. 3 strands measures 11.2–14.3 ppm, 8.8–11.8 ppm and 8.5–13.1 ppm, respectively, before optimization, while it is 8.8–11.8 ppm, 9.2–10.8 ppm and 7.9–11.6 ppm, respectively, after optimization. The oxygen content after optimization is lower than that before optimization, especially in the first and third strands. This is because, as mentioned above, the stagnation time and average residence time of molten steel in tundish after optimization are both improved, which provides sufficient time for the inclusions in molten steel to float. In addition, the consistency among strands significantly improves due to the more uniform flow field after optimization.

The as-cast pieces of blooms after acid etching were sampled by drilling with a φ5 mm drill for chemical analysis as shown in Figure 9. A total of 25 points were drilled in each bloom from No. 1 to No. 25, and their carbon contents were measured by an HCS-140 infrared carbon and sulfur meter. The local carbon segregation index was calculated based on the ratio of its carbon content to the average carbon content at all points, namely: C*_i_*/(∑C*_i_*/25), where *i* = 1, 2, …… 25. Three heats were taken respectively before and after optimization, and each heat took three strands of blooms. Therefore, there are 18 pieces (3 heats × 3 strands × 2 working conditions = 18) in total for the statistics. The C segregation indexes of all points are shown in Table 9, and those along the cross-section center line (from points 1 to 9) are shown in Figure 10.

As can be observed in Table 9, the carbon segregation indexes in equiaxed crystal and columnar crystal zones for different batches of bloom pieces fluctuates between 0.95 and 1.05. There seems to be no clear change before and after optimization, but the consistency among different strands is significantly improved as shown in Figure 10, and the segregation index of the center point 5 in No. 2 strand is also reduced from 1.13 to 1.10. This is because the element segregation is relevant to the as-cast structure, which is closely related to the superheat of molten steel [26]. After optimization, the temperature distribution of molten steel in tundish is more uniform as shown in Figure 6. The superheat for each strand casting is at the same level, and thus their solidification and segregation consistency are significantly improved accordingly, together with a greatly reduced center carbon segregation index.

### 5.4. Comparison on the Quality of Rolled Products by Overall Stactistical Data

The mechanical properties of hot-rolled products before and after tundish optimization, including tensile strength Rm, yield strength Rel, elongation A and reduction of area Z, were measured using a universal material testing machine WAW-Y500C, and the hardness measured with an HB-3000B Brinell hardness tester. Additionally, the ferritic phases and band structures were observed using a Carl Zeiss optical microscope after samples were etched with 2% nitric acid alcohol solution. In addition, the grain sizes were evaluated by the standard method with water-quenched samples after insulating at 860 °C for one hour. Finally, the product quality indexes measured above are comparatively listed in Table 10.

According to the big data analysis, after the flow field of the tundish is optimized, the pass rate of tensile strength of rolled products is 98.9%, which is higher than 98.3% before optimization, while the yield strength, elongation and section shrinkage show no clear change. The hardness fluctuation (maximum value minus minimum value) decreases from 84 HBW before optimization to 60 HBW after optimization. The ferrite content is stabilized from 3–9% to 3–8%. The band structure decreases from 2.08 to 1.87 grade. Inclusions B + C + D + Ds decrease from 1.143 to 1.105 grade, and the occurrence rate of Ds inclusions decrease from 0.192% to 0.118%. Ds refers to single particle inclusion, which are large in size and harmful to steel service property, thus requiring strict control in many special steels. In this study, the occurrence rate of Ds-type inclusions in C60S6 steel is reduced by the optimization of tundish, since the stagnation time and average residence time of strands 1 and 2 are prolonged, and the short-circuit flow is eliminated after optimization, which gives the inclusions adequate time to float and be removed. In addition, as mentioned above, the optimized tundish structure can alleviate the lining erosion in zone A and thus reduce large-sized exogenous inclusions into the molten steel. The A-type inclusion was not counted in this study because the steel contains S and no conspicuous related service defects in its rod products from A-type inclusions. Moreover, no clear grain-size index variation was observed in this study.

### 5.5. Products Customer Feedback

According to user feedback regarding the cracking con-rod products, the failure rate of products was controlled within 4‰ during their forming processes, and the fracture of connecting rod was relatively smooth compared with that before casting practice optimization.

## 6. Conclusions

The flow field optimization to a multi-strand asymmetric tundish has been conducted through water modelling combined with a numerical calculation method for improved metallurgical behavior, and the beneficial results were revealed accordingly through volume production. The conclusions are drawn as follows: (1)The water modeling experiments show that dead zone proportion of the tundish is reduced from 53.08% originally to 21.10% after optimization, and the stagnation time of its outlets 1 and 2 are extended to about twice that of before, indicating clear improvement in the consistency among strands. The average residence time of the three strands measures 591 to 651 s, much longer than the 267 to 434 s in the original case. This will provide sufficient time for inclusions in the molten steel to float and be removed, and thus improve the cleanliness of steel.(2)The numerical simulation results show that the flow field in the tundish is more uniform due to the adoption of a vortex inhibitor and an optimized wall structure, which is beneficial for temperature-field uniformity. The maximum temperature difference at the three outlets of the tundish is only 1 K, 2 K lower than the original case. The maximum temperature difference in the whole outlet vertical section is decreased from 8 K to 6 K and both the high and low temperature areas are reduced by the optimized case.(3)The maximum temperature difference of molten steel measured among the outlets around stoppers of the three strands is decreased from an initial 2–5 °C to 1–3 °C after optimization; the lining erosion in region A near the No. 1 strand of the tundish is clearly alleviated due to the disappearance of the local short-circuit flow.(4)The differences in columnar crystal rate among different strands of the bloom castings are decreased from 2.27–3.17% to 1.26–1.85% after tundish optimization; the consistency of the strand central line carbon segregation index is significantly improved. The total oxygen content difference among strands is reduced from 1.7–3.5 ppm to 0.8–1.9 ppm, with the No. 1 strand showing a more significant effect than the other two strands.(5)The main mechanical properties and microstructure stability of the rolled products are statistically well improved owing to the generally identical as-cast primary quality, in which the pass rate of tensile strength reaches 98.9%, the hardness dispersion is decreased from 84 HBW to 60 HBW, the non-metallic inclusion grade of B + C + D + Ds is decreased from 1.143 to 1.105, and the occurrence rate of Ds type inclusion is decreased from 0.192% to 0.118%. Additionally, the band-structure level of the hot-rolled products drops from 2.08 to 1.87.

In summary, flow-field optimization of the tundish reveals a clear beneficial effect to its subsequent castings and hot-rolled products, especially for an improved identical quality index while cast by different strands. 

## Figures and Tables

**Figure 1 materials-15-03698-f001:**
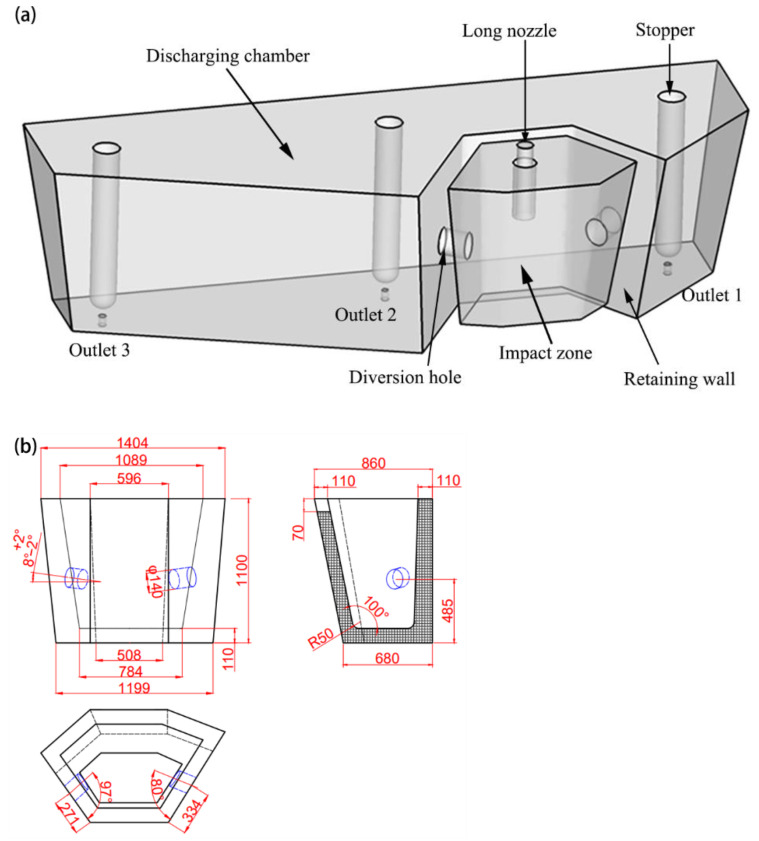
Schematic diagram of the prototype tundish (**a**) and flow control device (**b**), unit: mm.

**Figure 2 materials-15-03698-f002:**
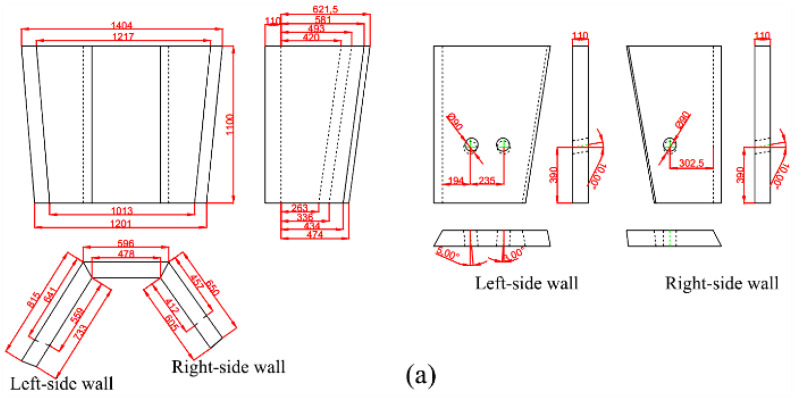
Retaining wall structure (**a**) and vortex inhibitor (**b**) in optimized case B and their geometry sizes, unit: mm.

**Figure 3 materials-15-03698-f003:**
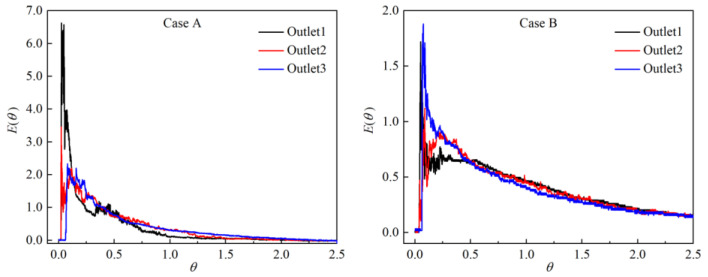
Comparison of RTD curves between cases A and B.

**Figure 4 materials-15-03698-f004:**
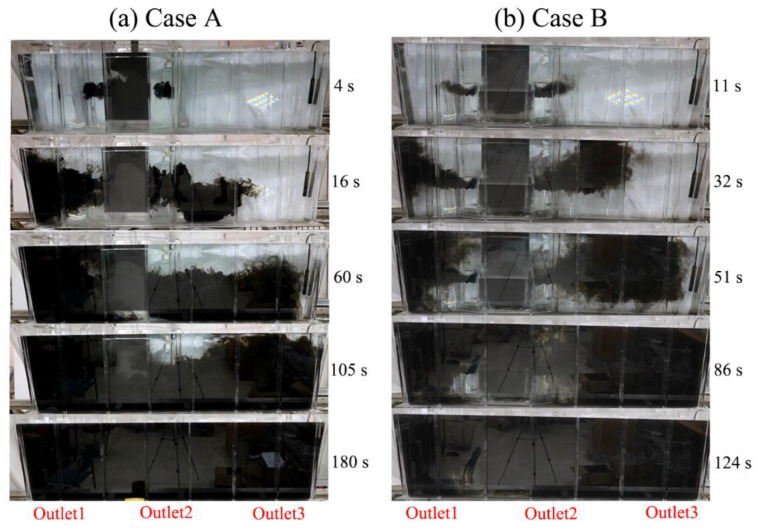
Ink trajectory comparison of cases A (**a**) and B (**b**).

**Figure 5 materials-15-03698-f005:**
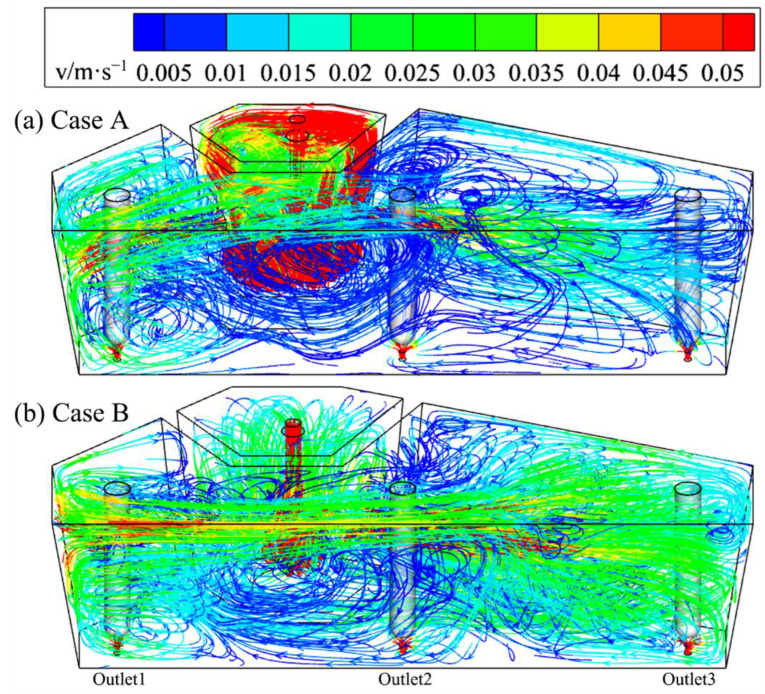
Three-dimensional streamline contours of tundish in cases A (**a**) and B (**b**).

**Figure 6 materials-15-03698-f006:**
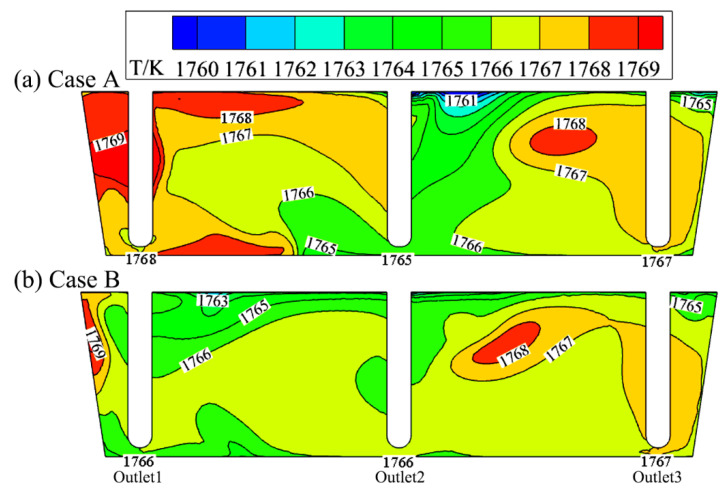
Comparison of temperature distribution of cases A (**a**) and B (**b**) in the vertical section of outlet center.

**Figure 7 materials-15-03698-f007:**
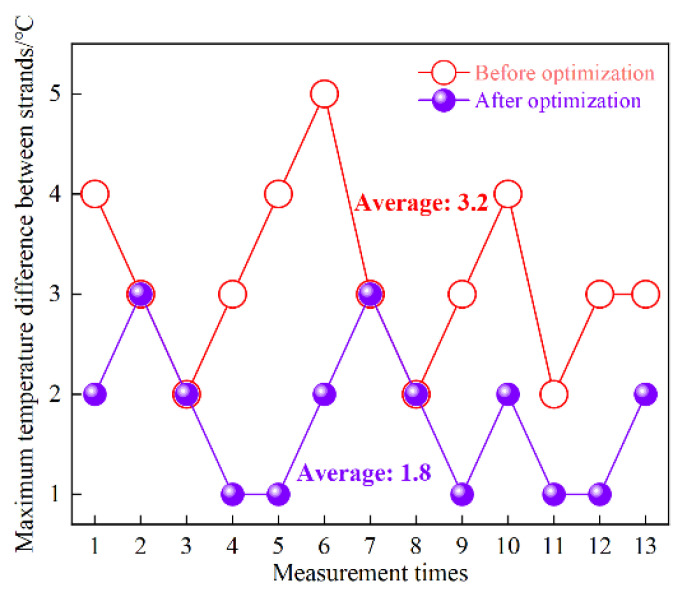
Maximum temperature difference among strands before and after tundish optimization.

**Figure 8 materials-15-03698-f008:**
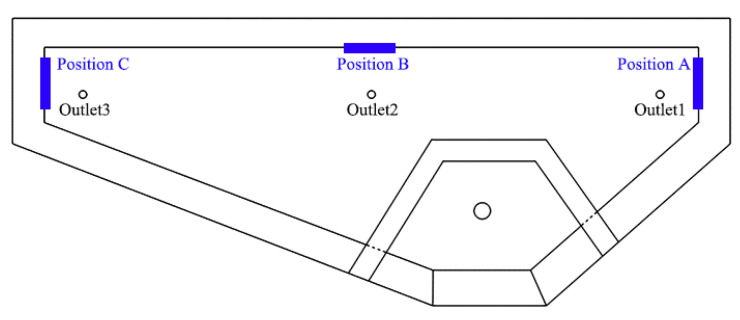
Schematics of erosion examination locations.

**Figure 9 materials-15-03698-f009:**
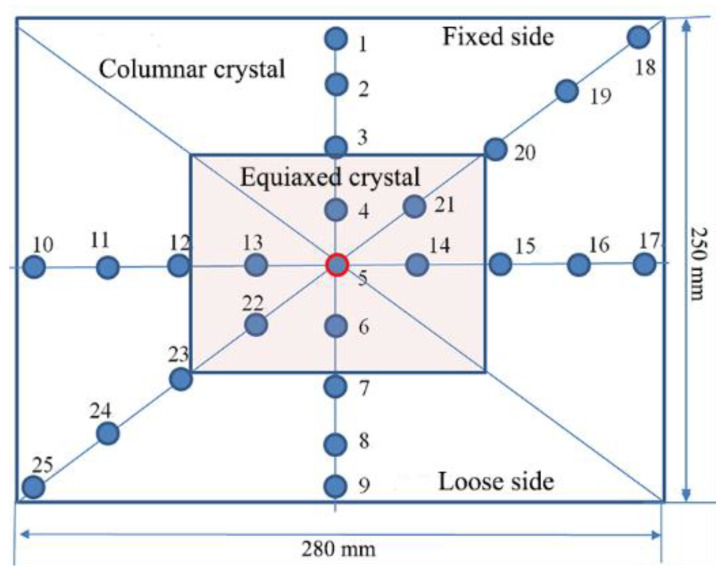
Schematic diagram of sampling location for C segregation measurement.

**Figure 10 materials-15-03698-f010:**
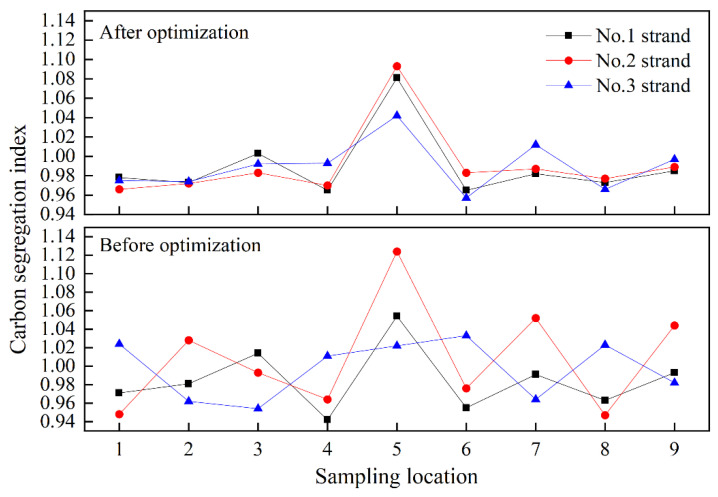
Carbon segregation index before and after optimization.

**Table 1 materials-15-03698-t001:** Chemical composition of steel C70S6, wt%.

C	Si	Mn	P	S	Cr	Ni	Mo	Al	V	N
0.67–0.73	0.15–0.25	0.53–0.60	≤0.045	0.060–0.070	0.10–0.15	0.04–0.08	≤0.03	≤0.010	0.030–0.040	0.012–0.016

**Table 2 materials-15-03698-t002:** Process parameters for the continuous casting of C70S6 steel.

Parameter	Value
Tundish capacity	21 t
SEN inner diameter	40 mm
Inner diameter of ladle nozzle	85 mm
Depth of the molten bath in tundish	950 mm
Section size of the bloom caster	250 mm × 280 mm
Casting speed	0.75 m/min

**Table 3 materials-15-03698-t003:** Calculation formula of flow characteristic parameters of molten steel in tundish.

Parameters	Symbol and Unit	Formula
Theoretical mean residence time	$t_{f}$, s	$V_{m}/Q_{m}$
Actual average residence time	$t_{\mathrm{av}}$, s	$\int_{0}^{\infty }tC(t)dt/\int_{0}^{\infty }C(t)dt$
Volume fraction of plug flow	$\theta_{p}$, %	$\frac{1}{2}(t_{\min}+t_{\mathrm{peak}})/t_{f}$
Volume fraction of dead zone	$\theta_{d}$, %	$1-t_{\mathrm{av}}/t_{f}$
Volume fraction of well-mixed zone	$\theta_{m}$, %	$1-\theta_{p}-\theta_{d}$
Standard deviation between strands	*S*, s	$\sqrt{\left[\sum_{j=1}^{n}{\left(x_{j}-\bar{x}\right)}^{2}]/(n-1\right)}$

**Table 4 materials-15-03698-t004:** Governing equations of the mathematical model.

Governing Equations	Symbols
1. Continuity equation $\frac{\partial (\rho u_{i})}{\partial x_{i}}=0$	$\rho \;\mathrm{is}\;\mathrm{density}\;\mathrm{of}\;\mathrm{molten}\;\mathrm{steel},\;u_{i}$$\;\mathrm{is}\;\mathrm{fluid}\;\mathrm{velocity},\;\mu_{eff}$ is effective viscosity, *P* is pressure.
2. Momentum conservation equation $\frac{\partial (\rho u_{i}u_{j})}{\partial x_{j}}=-\frac{\partial p}{\partial x_{i}}+\frac{\partial }{\partial x_{i}}[\mu_{eff}(\frac{\partial u_{i}}{\partial x_{j}}+\frac{\partial u_{j}}{\partial x_{i}})]+\rho g_{i}$	*x* is the spatial coordinates at *i* and *j* directions.
3. Turbulent equations $\frac{\partial (\rho ku_{i})}{\partial x_{i}}=\frac{\partial }{\partial x_{i}}\left[(\mu_{eff}+\frac{\mu_{t}}{\sigma_{k}})\frac{\partial k}{\partial x_{i}}\right]+G-\rho \varepsilon$ $\frac{\partial (\rho \varepsilon u_{i})}{\partial x_{i}}=\frac{\partial }{\partial x_{i}}\left[(\mu_{eff}+\frac{\mu_{t}}{\sigma_{\varepsilon }})\frac{\partial \varepsilon }{\partial x_{i}}\right]+C_{1}\frac{\varepsilon }{k}G-C_{2}\rho \frac{\varepsilon^{2}}{k}$ $G=\mu_{t}\frac{\partial u_{j}}{\partial x_{i}}(\frac{\partial u_{i}}{\partial x_{j}}+\frac{\partial u_{j}}{\partial x_{i}})$ $\mu_{eff}=\mu +\mu_{t}=\mu +\rho C_{\mu }\frac{k^{2}}{\varepsilon }$ $C_{1}=1.44$ $,\;C_{2}=1.92$ $,\;C_{\mu }=0.09$ $,\;\sigma_{k}=1.0$ $,\;\sigma_{\varepsilon }=1.3$	*k* is turbulence kinetic energy, *ε* is the turbulent kinetic energy dissipation rate, *σ_k_* and *σ_ε_* are the Schmidt numbers for *k* and *ε*, *G* is generation rate of turbulence energy and *μ_eff_* is effective viscosity.
4. Energy equation $\frac{\partial }{\partial x_{i}}(\rho Tu_{i})=\frac{\partial }{\partial x_{i}}\left(K_{eff}\frac{\partial T}{\partial x_{i}}\right)$	$T\;\mathrm{is}\;\mathrm{temperature},\;K_{eff}$ is effective thermal conductivity.

**Table 5 materials-15-03698-t005:** Boundary conditions and material parameters.

Boundary Condition and Material Parameter	Value	Boundary Condition and Material Parameter	Value
Longitudinal wall heat flux, W/m^2^	−4600	Density of liquid steel, kg/m^3^	7014
Transversal wall heat flux, W/m^2^	−4000	Specific heat of liquid steel, J/(kg·K)	750
Free surface heat flux, W/m^2^	−15,000	Thermal conductivity of liquid steel, W/(m·K)	41
Bottom heat flux, W/m^2^	−1800	Viscosity of liquid steel, kg/(m·s)	0.0061
*k*, m^2^/s^2^	0.00238	*ε*, m^2^/s^2^	0.00274

**Table 6 materials-15-03698-t006:** Flow characteristic parameters of the two experimental schemes.

Case	Outlet No.	*t*_min_/s	*t*_peak_/s	*t*_av_/s	*θ*_d_/%	*θ*_p_/%	*θ*_m_/%	*S*_tmin_/s	*S*_tav_/s
A	1	17	20	267	53.08	2.38	44.53	19.20	25.56
	2	16	19	369					
	3	50	64	434					
B	1	30	41	651	21.10	5.59	73.31	11.12	15.22
	2	31	71	637					
	3	50	60	591					

**Table 7 materials-15-03698-t007:** Mesh independence study and model validation.

Case	Number of Grids	*T*_outlet1_/K	*T*_outlet2_/K	*T*_outlet3_/K	Maximum Temperature Difference among Three Strands for Industrial Measurement/°C
A	309,506	1768	1765	1767	2–5, average 3.2
	394,471	1768	1765	1767	
	489,901	1768	1765	1767	
	740,025	1769	1765	1766	
	1,093,481	1768	1767	1765	
B	321,107	1766	1766	1767	1–3, average 1.8
	431,562	1766	1766	1767	
	559,800	1766	1766	1767	
	736,498	1766	1766	1767	
	1,031,898	1766	1767	1767	

**Table 8 materials-15-03698-t008:** Erosion state of the working layer in tundish.

Item		Before Optimization	After Optimization
Measured number of tundish, #		3	5
Erosion width, mm	A	90–120	90–100
	B	90–110	90–110
	C	90–110	90–110
Erosion depth, mm	A	10–12	9–11
	B	10–12	9–11
	C	10–12	9–11

**Table 9 materials-15-03698-t009:** Comparison of C70S6 bloom quality before and after tundish optimization.

Item	Working Condition		Before Optimization					After Optimization				
	Detected Sample No.		1#	2#	3#	4#	5#	1#	2#	3#	4#	5#
As-cast macrostructure	Columnar crystal rate, %	No. 1	49.25	46.27	56.1	48.75	53.81	47.80	52.56	51.00	49.96	53.88
		No. 2	48.56	43.86	58.19	46.36	51.50	49.06	53.50	52.85	48.75	55.17
		No. 3	46.08	46.48	55.92	46.87	54.15	48.70	54.25	51.35	48.54	54.86
		Δ_max_	3.17	2.62	2.27	2.39	2.65	1.26	1.69	1.85	1.42	1.29
	Equiaxed crystal rate, %	No. 1	28.32	25.04	22.21	28.71	27.96	30.36	25.40	24.38	26.21	20.06
		No. 2	28.41	26.84	24.55	27.91	28.13	30.17	26.36	23.28	26.43	21.72
		No. 3	30.87	27.74	23.66	30.78	30.21	29.88	27.24	24.49	27.81	19.62
		Δ_max_	2.55	2.70	2.34	2.87	2.25	0.48	1.84	1.21	1.60	2.10
	Central shrinkage	No. 1	1.5	1.0	1.5	1.0	1.5	1.5	1.5	1.5	1.5	1.0
		No. 2	1.5	1.0	1.5	1.0	1.5	1.5	1.5	2.0	1.0	1.0
		No. 3	1.5	1.0	1.5	1.5	2.0	1.5	1.5	1.5	1.5	1.5
		Δ_max_	0.0	0.0	0.0	0.5	0.5	0.0	0.0	0.5	0.5	0.5
C segregation index at different regions of bloom	Columnar crystal zone	No. 1	0.97–1.03	0.98–1.03	0.98–1.08	-	-	0.97–1.03	0.96–1.04	-	-	0.98–1.03
		No. 2	0.96–1.04	0.95–1.01	0.95–1.03	-	-	0.96–1.02	0.96–1.02	-	-	0.96–1.03
		No. 3	0.95–1.02	0.94–1.03	0.99–1.02	-	-	0.99–1.02	0.98–1.01	-	-	0.96–1.02
	Equiaxed crystal zone	No. 1	0.95–1.04	0.93–1.02	0.96–1.03	-	-	0.95–1.03	0.96–1.04	-	-	0.95–1.03
		No. 2	0.97–1.05	0.96–1.03	0.95–1.01	-	-	0.97–1.01	0.98–1.03	-	-	0.97–1.02
		No. 3	0.95–1.03	0.98–1.03	0.97–1.03	-	-	0.96–1.03	0.95–1.02	-	-	0.96–1.3
	Center point	No. 1	1.11	1.05	1.07	-	-	1.10	1.09	-	-	1.08
		No. 2	1.03	1.12	1.12	-	-	1.09	1.03	-	-	1.09
		No. 3	1.05	1.02	1.03	-	-	1.10	1.08	-	-	1.04
T. O.	ppm	No. 1	12.3	14.3	13.2	11.8	11.2	10.8	8.8	11.2	9.9	11.8
		No. 2	8.8	11.8	10.4	9.4	9.9	9.2	9.2	10.8	10.2	10.7
		No. 3	10.2	10.9	13.1	8.5	9.5	8.9	7.9	11.6	10.8	10.5
		Δ_max_	3.50	3.40	2.80	3.30	1.70	1.90	1.30	0.80	0.90	1.30

**Table 10 materials-15-03698-t010:** Comparison of the quality of rolled products before and after tundish optimization.

Item		Before Optimization				After Optimization			
Mechanical property	Number of samples, #	1153				912			
	Measured item	Rm, MPa	Rel, MPa	A, %	Z, %	Rm, MPa	Rel, MPa	A, %	Z, %
	Standard requirement	900–1050	>550	>10	20	900–1050	>550	>10	20
	Measured value	878–1050	485–599	8.5–16	13–29	849–1018	439–692	6.5–19.5	8–48
	Average value	943	555	12.7	21.5	941	552	11.9	20.5
	Pass rate, %	98.3	99.6	97.8	98.5	98.9	99.6	97.9	98.4
Hardness	Number of samples, #	1095				875			
	Measured value, HBW	229–313				225–285			
	Average value, HBW	258				256			
	Hardness difference (Maximum minus minimum), HBW	84				60			
Microstructure	Number of samples, #	70				93			
	Microstructure	F + P				F + P			
	Ferritic content, %	3–9				3–8			
Grain size	Number of samples, #	551				448			
	Measured value	7.0–8.0				7.0–8.0			
Band structure index	Number of samples, #	54				56			
	Measured value	1.0–2.5				1.0–2.5			
	Average value	2.08				1.87			
Inclusions	Number of samples, #	1044				848			
	Grade of (B + C + D + Ds)	1.143				1.105			
	Ds occurrence rate, %	0.192				0.118			

## Data Availability

The data presented in this study are available on request from the corresponding author.

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
