# Peer review of "Effect of Flow Field Optimization of an Asymmetric Multi-Strand Tundish on the Quality Consistency of Cracking Con-Rod Steel"

_materials, 2022, doi:10.3390/ma15103698_

Round 1

Reviewer 1 Report

The study is quite interesting and has a different perspective. but it is useful to make some corrections in some points.
1. Why the geometry in figure.1 was chosen should be explained in more detail. The visual can be made more beautiful with schematic drawings instead of technical drawings.

2. In addition, the font sizes in table.2 and 3 should be reduced.

3. It should be stated on what basis the values specified in Table.5 were taken, and references can be made if possible.

4. Information should be given about the ink used in Line 247, and its behavior for both cases should be examined in more detail.

5. Significant differences in numeric analysis results shown in Figure.5, why did they occur? should be supported by literature.

Reviewer 2 Report

  • The abstract looks good.Please include all significance numerical results.

  • Scientific citation numbers are not enough. For the introduction, please add more current references and briefly explain them.

  • In the last paragraph of the introduction, it should be expressed the novelty of the study, the differences from the past in detail.

  • There is no experimental validation for numerical results. Moreover, there is no mesh independent tests. These are very important for numeric analysis. Please provide them.

  • Results and discussion and conclusion parts are inadequate according to citation and analyze in detail. There should be the importance of the study in detail, comparison results with other approaches in literature, the success of the experimental and computational results.

  • Improve the results and discussion and conclusion parts.

  • Please fix the typographical and eventual language problems in paper.

  • The paper is well-organized yet there is a reference problem. First, your reference list contains one paper from “Materials” journal. If your work is convenient for this journal’s context then there are many references from this journal. Secondly, cited sources should be primary ones. Namely, indexed area shows the power of a paper and directly your paper’s reliability. Please make regulations in this direction.

  • The article should be rearranged by taking into account the journal writing rules and citation rules.

*** Authors must consider them properly before submitting the revised manuscript. A point-by-point reply is required when the revised files are submitted.

Round 2

Reviewer 2 Report

The parts that needed to be revised were corrected by the authors. I would like to state that it is now ready to be published